# Selection Factors Influencing Eventual Owner Satisfaction about Pet Dog Adoption

**DOI:** 10.3390/ani12172264

**Published:** 2022-09-01

**Authors:** Ian R. Dinwoodie, Vivian Zottola, Karla Kubitz, Nicholas H. Dodman

**Affiliations:** 1Cummings School of Veterinary Medicine, Tufts University, North Grafton, MA 01536, USA; 2Center for Canine Behavior Studies, Salisbury, CT 06068, USA; 3Boston K9 Concierge LLC, Boston, MA 02127, USA; 4Department of Kinesiology, Towson University, Towson, MD 21204, USA

**Keywords:** dogs, adoption, survey, behavior, expectations

## Abstract

**Simple Summary:**

Pet dog adopters are influenced by a variety of complex factors some of which are ethical, emotional, and humanitarian, including wanting companionship for themselves or other pets, the dog’s breed, age, appearance, temperament, or behavior, or to provide a home for a homeless dog. However, not all adoptions are successful and managing owner expectations preadoption is difficult to navigate. Using a self-reporting questionnaire, we found that consideration of a dog’s personality and behavior had a positive effect on eventual owner satisfaction. Owners who adopted a dog for companionship were more likely to be satisfied than owners compelled by any other motive. In addition, less forethought prior to adoption, ideally less than one week, was found to increase the likelihood of eventual owner satisfaction. We suggest that consideration of these factors prior to adoption may lead to more successful pet dog adoption outcomes.

**Abstract:**

Personal likes, experience, and deep-rooted interests to satisfy emotional needs such as companionship, affection, empathy, and security are some of the underlying human motivations for acquiring a pet companion. In this study, we asked how long the owner took to decide whether to adopt a dog, who their dog was adopted from, their primary motivation for adoption, a ranking of characteristics considered during the adoption process, and how satisfied they were with the eventual outcome. Participants (*n* = 933) to this Center for Canine Behavior Studies survey completed an online questionnaire with responses representing 1537 dog/owner pairs. A majority of participants reported satisfaction with at least one of their adopted dogs. Odds of eventual satisfaction are higher for participants who spent less than a week considering an adoption or were seeking a pet to provide companionship and affection. Participants that prioritized personality as an adoption criteria were more likely to be satisfied with their adopted dogs. A mast majority (91%) of participants reported they would consider adopting another dog in the future. Selection criteria rankings that participants indicated they would employ for future adoptions tended to shift away from physical to behavior characteristics when compared to selection criteria priorities of prior adoptions.

## 1. Introduction

In the United States, most people acquire pet dogs from various distribution channels including commercial and backyard breeders, pet stores, rescues, and shelters where dogs may be selected in person or online [1,2]. Whatever the source, when making decisions adopters are influenced by a variety of complex factors some of which are ethical, emotional, and humanitarian, including wanting companionship for themselves or other pets, the dog’s breed, age, appearance, temperament, or behavior, or to provide a home for a homeless dog [1,3]. Other motivations for acquiring dogs satisfy more functional needs such as wanting an exercise and adventure partner, enhanced social interactions with other people, needing protection, or seeking an assistance or therapy dog [4]. If the qualities a prospective owner is looking for, such as a particular temperament, are not met the failure of expectations can affect adopted dogs’ later welfare and even survival [5]. Veterinarians in private practice deal with clients closely, answering questions, discussing training methods, and helping with problems and difficulties before or after they arise [6]. Kidd et al. [6] found that owners of adopted dogs who avail themselves of veterinary advice are less likely to relinquish their adopted charge. In another study, Kidd et al. [7] found that greater adoption success was attained with owners who had previously owned pets, that men rejected a significantly higher percentage of pets than women, parents rejected more pets than non-parents, and that specific expectations differed considerably between men and women, parents and nonparents, and retainers and rejecters of adopted pets.

Dogs acquired from breeders or pet shops are generally chosen because the new owner wants a particular breed, though other factors, such as appearance, age, size, and temperament or behavior, also factor into the decision [8]. Humanitarian motivation features strongly in the adoption of rescue dogs. Rescue dogs are dogs of any breed or age that have been abandoned or relinquished and placed in a temporary boarding system (“shelter”). These dogs may have been found as strays or simply been surrendered by their previous human caretakers for a plethora of reasons. Rescue dogs can be successfully adopted from shelters, though some are subsequently brought back to the shelter if the new owner is disappointed or dissatisfied [9,10,11].

Not all adoptions are successful and managing owner expectations preadoption is difficult to navigate. If owner expectations are high and skewed based on limited knowledge, they may be difficult to achieve and therefore returns are more likely [12]. Well respected scientists have attempted to understand this phenomenon and investigate underlying motivating factors influencing long term success in pet dog adoptions. For example, personality—both dog and human—alone cannot determine successful outcome of adoptions [13]. Ensuring compatibility with all household members prior to adoption was looked at and was found to improve rehoming success rates [14], as was providing training post adoption [15]. In assessing the personality of 88 dogs and their owners from Oklahoma, Curb et al. [16] found four correlations between owner satisfaction scores and dog–owner personality match; these were (1) the tendency to share possessions, (2) love of running outside, (3) likeliness of being destructive, and (4) ability to have a good relationship with others. The authors suggest that prospective dog owners may want to consider adopting dogs that match their personality on these characteristics. In a similar Danish study of 421 dog-owner pairs, it was found that owner characteristics appeared to influence the dog-owner relationship more than dog personality traits [17]. The results of these two latter studies, while informational, require that owners assess their own personality and match it to the prospective adoptee dog to increase the odds of eventual owner satisfaction. Neither study found dogs’ physical appearance, breed, signalment, personality/behavior, or trainability to be a helpful predictor of owner satisfaction, though the study by Curb et al. [16] alluded to compatibility with other pets, destructiveness, and a love of exercise as potentially helpful factors to consider when adopting a dog. Dog personality/behavior was loosely represented by destructiveness (presumably the lack thereof) and the love of exercise in the latter study. In another study of owned dogs, van Herwijnen at al. [18] found that increasing levels of aggressive behaviour significantly lowered the chance of owners being very satisfied with their dog. This study also found that disobedience was associated inversely with ownership satisfaction to a similar degree, possibly because unwanted behaviour by the dog, including aggression, was found to coincide with high perceived costs, including re-training expenses.

Discerning baseline behavior of shelter dogs is difficult as depending on the individual dog and their learned history, these may be suppressed while living in the shelter system and only presenting post adoption when living in a new home environment. In many cases post adoption may be too late [19]. A one-year cohort study including 14 centers conducted in the United Kingdom found 14% of adopted dogs were returned to shelters within six months of adoption [20]. Other studies found return periods as short as hours to days [21]. Whether an adopter returns a dog to a shelter has been shown to depend on the severity of the behaviors presented as well as the training provided post adoption. Untreated behaviors have been shown to harm human pet dog relations and, in some cases, quickly. This is likely due to an underestimated commitment of time, money and emotional investment required without guarantees [20,22].

To understand consumer interests and improve matching efforts when adopting dogs, the Center for Canine Behavior Studies, Inc 501(c)(3) (CCBS) members and general public were asked to provide feedback by an online qualitative survey. In this study, involving 933 participants, we examined what factors consumers prioritized when acquiring a pet dog and their subsequent level of satisfaction with the owner/dog relationship, while other studies have been conducted to examine factors influencing potential adopters’ preferences [23,24,25], none were designed to examine the result of selection criteria on the eventual outcome success of the adoption process. Correlation of selection factors on the eventual outcome was the purpose of the present study.

## 2. Materials and Methods

### 2.1. Data Collection

The questionnaire for this study was developed and hosted on Typeform, an online survey service platform. A link to the public questionnaire was posted on social media platforms (Twitter, Instagram, and Facebook) and distributed to CCBS members via email. Data collection spanned one year starting from 13 February 2020. Participants were willing dog owners who voluntarily completed the online questionnaire. All data in this study was self-reported by participants (e.g., primary motivation is owner self-reported primary motivation).

Logically, the distributed questionnaire could be considered in three parts: (1) demographic information about the owner, (2) information about a single dog that was adopted (i.e., taken into a relationship by choice), and (3) information about future dog adoption. Participants with more than one dog were prompted to fill out the second part of the questionnaire for each of their dogs. The owner demographics component of the questionnaire was used to gather the age and gender of the owner. The individual dog component of the questionnaire was used to gather the length of time spent thinking about an adoption (choices: less than one week, one week to six months, greater than six months to less than six years, or greater than six years), the owner’s primary motivation for the adoption (choices: companionship/affection, social interaction, exercise/adventure partner, protection, someone else in the house wanted a dog, or companion for another pet), the adoption source (choices: breeder, online pet shop, local pet shop, shelter/rescue, family member/friend, found, or another country/island), the ranking of characteristics considered for the adoption, if the adopted dog was currently living with the owner, and whether the owner’s expectations for that dog had been met (choices: yes, partially, or no). The characteristics that owners were asked to rank were: age, appearance, breed, compatibility with other pets, personality, size, and trainability. Two additional questions were presented to owners when they indicated they were not living with the adopted dog: how long the owner had lived with the dog (choices: less than one week, one week to six months, greater than six months to less than six years, or greater than six years) and the living situation of the dog (choices: rehomed, surrendered, euthanized, lost, or passed). The future adoption component of the questionnaire was used to gather whether the owner would consider adopting a dog again in the future and, if so, their revised ranking of characteristics for a future adoption. The characteristics presented to users for ranking were identical to the characteristics from the previous component.

Email addresses provided by users were recorded as randomly generated MD5 checksums (i.e., 128-bit hashes) to avoid retaining any personally identifying information. Responses across the three parts of the questionnaire were linked using these hashes. As a result, participants remained anonymous to all personnel involved and no ethical approval was required for undertaking this study.

An annotated copy of the questionnaire used for study is available in Appendix A.

### 2.2. Data Preprocessing

The study data set was tidied using the R programming language (version 4.2.0) provided by the R Foundation for Statistical Computing [26] and packages from the tidyverse library [27], namely the dplyr [28], tidyr [29], forcats [30], and purrr [31] packages. Repeat responses were those that had non-unique owner/dog pairs. Of the repeat responses, only the most recent response was retained for the study. For each submitted questionnaire the ranked list of characteristics was exploded such that each characteristic was given its own column and assigned a value corresponding to its position in the list (i.e., the first ranked characteristic was assigned a value of 1 and the last ranked characteristic was assigned a value of 7). The local pet store and online pet store responses categories were collapsed down to a single local/online pet store category to minimize unnecessary complexity.

### 2.3. Inclusion Criteria

Only responses from adults, defined as participants with a reported age of 18 years or older, were retained for analysis. Complete responses were those that included data considered necessary to build relevant statistical models: owner demographics information, the length of time spent thinking about an adoption, the owner’s primary motivation for the adoption, the adoption source, and the ranking of characteristics considered for the adoption. Incomplete responses were excluded from the study sample.

### 2.4. Descriptive Analysis

The study data set was exported from Typeform as a comma-separated values (CSV) file. All analyses were performed using the R programming language (version 4.2.0) provided by the R Foundation for Statistical Computing [26]. Descriptive statistics were calculated. Ranges were provided for all medians. Nonreduced proportions were provided for all percentages for clarity. Graphics were created using methods provided by the ggplot2 package [32].

### 2.5. Analytical Modeling

The significance level was set to *α* ≤ 0.05 for all regression models in this manuscript. Corrections for multiple testing were performed using the Benjamini and Hochberg adjustment [33].

#### 2.5.1. Change in Rankings

Mann–Whitney tests were performed using methods from the stats package [26] to evaluate the difference between past and future adoption characteristic rankings. The null hypothesis was that there was no difference.

#### 2.5.2. Impact of Adoption Criteria

A hierarchy of binary logistic regression models was built to assess the impact of factors influencing adoption on adoption satisfaction. An owner was considered satisfied with an adoption if they had indicated that the dog had fully met their expectations. A null model (i.e., intercept only) was built to serve as a frame-of-reference. The first, second, and third steps of the hierarchy added terms to account for background variables, adoption source, and primary adoption motivation, respectively. Background variables consisted of the owner’s age, gender, and time spent thinking about the adoption. Each model resulting from this process was compared with its preceding model.

Due to the compositional nature of ranked data (i.e., the constant-sum constraint across a complete set of rankings), models were limited to the inclusion of a single characteristic rank term. To evaluate all characteristics, the fourth and final step of the hierarchy was broken into seven parallel models; each model added a single characteristic rank term to the model from the third step of the hierarchy.

Binary logistic regression models were built using the generalized linear model (GLM) fitter provided by the stats package [26]. Two-way contingency tables were constructed for categorical predictors to verify a minimum of ten responses for the least frequent outcome (i.e., the “rule of ten”) [34]. Odds ratios (ORs) were calculated as a measure of effect size. Confidence intervals were calculated using the bootstrapped samples with replacement (*N* = 10,000) using methods provided by the rsample package [35]. Multicollinearity was assessed by variance inflation factor (VIF) using methods provided by the car package [36]. Overall evaluation was performed for each model via likelihood ratio test. Fit of the models against actual outcomes was assessed using the Hosmer–Lemeshow goodness-of-fit test (H–L) [37] implemented by the performance package [38]. Pseudo-*R*^2^s were calculated using methods proposed by Nagelkerke [39] and implemented by the performance package [38] to provide an additional measure of goodness-of-fit.

## 3. Results

### 3.1. Demographics

A total of 1595 responses were provided by 971 participating dog owners. Four percent (*n* = 58/1595) of the responses were dropped due to their failure to meet the inclusion criteria. The resulting sample study consisted of complete responses for 1537 owner/dog pairs across 933 participants.

Ninety-two percent (*n* = 859/933) of the participating dog owners were female; the remainder (*n* = 74/933) were male. Participants averaged an age of 51 years old (range: 18 to 85 years old). Female participants averaged 52 years old (range: 18 to 85 years old) while male participants averaged 58 years old (range: 20 to 75 years old). The median number of dogs per household was one (range: 1 to 10 dogs per household). More than half (*n* = 495/933) of the participants represented single dog households. Ninety-one percent (*n* = 851/933) of owners indicated that they would consider another dog adoption in the future, seven percent (*n* = 61/933) indicated that they would not consider dog adoption again in the future, and two percent (*n* = 21/933) elected not to provide a response.

Eighty-eight percent (*n* = 1347/1537) of the dogs lived with their participating owner. Of the minority subset of dogs that were not living with the participating owner, 75% (*n* = 142/190) had passed away, 18% (*n* = 34/190) were euthanized, four percent (*n* = 7/190) were not living with their owner for reasons other than those asked on the questionnaire, three percent (*n* = 5/190) had been rehomed, less than one percent (*n* = 1/190) had been lost, and less than one percent (*n* = 1/190) had been surrendered. Seventy-eight percent (*n* = 149/190) of the dogs not living with their participating owner had lived with their owner for more than six years, 19% (*n* = 36/190) for greater than six months but less than six years, two percent (*n* = 4/190) for one week to six months, and less than one percent (*n* = 1/190) for less than one week.

### 3.2. Time Spent Thinking

Forty-nine percent (*n* = 752/1537) of the dogs were acquired with one week to six months of forethought, 34% (*n* = 528/1537) with between six months and six years of forethought, 13% (*n* = 196/1537) with less than one week of forethought, and four percent (*n* = 61/1537) with greater than six years of forethought.

### 3.3. Adoption Source

Fifty-five percent (*n* = 847/1537) of the dogs were adopted from a rescue/shelter, 31% (*n* = 480/1537) from a breeder, eight percent (*n* = 130/1537) from friends or family members, four percent (*n* = 68/1537) from local or online pet shops, three percent (*n* = 51/1537) were found, and less than one percent (*n* = 12/1537) were adopted from a foreign country. Owners had an average age of 51 (range: 18 to 85 years old) for rescue/shelter adoptions, 56 (range: 19 to 84 years old) for breeder adoptions, 51 (range: 20 to 75 years old) for adoptions from friends or family members, 48 (range: 20 to 73 years old) for adoptions from a local or online pet shop, 49 (range: 22 to 74 years old) for owners of dogs that were found, and 53 (range: 36 to 65 years old) for adoptions from foreign countries.

### 3.4. Primary Motivation for Adoption

Fifty-five percent (*n* = 846/1537) of the dogs were acquired for companionship and affection, 15% (*n* = 224/1537) as a companion for another pet in the household, 12% (*n* = 187/1537) for reasons other than those explicitly asked for on the questionnaire, seven percent (*n* = 100/1537) for working or sporting, five percent (*n* = 77/1537) as an exercise and adventure partner, five percent (*n* = 70/1537) due to other members of the family wanting a dog, one percent (*n* = 21/1537) for social interaction, and less than one percent (*n* = 12/1537) were adopted for protection.

### 3.5. Characteristic Ranks

All characteristics, for both past and future adoptions, received rank positions that ranged from 1.00 to 7.00 (i.e., all characteristics had at least one response per minimum and maximum rank positions). Full distributions of ranks for each characteristic are displayed in Figure 1.

Characteristic rank means for past and future adoptions along with the difference between past and future rank means by characteristic are provided in Table 1. The change in rank position for each characteristic except for size was statistically significant.

### 3.6. Owner Satisfaction

Eighty-three percent (*n* = 1282/1537) of the adopted dogs fully met their owner’s expectations, 16% (*n* = 248/1537) partially met their owner’s expectations, and less than one percent (*n* = 7/1537) failed to meet their owner’s expectations. Ninety percent (*n* = 843/933) of owners indicated satisfaction with at least one dog and 76% (*n* = 707/933) of owners indicated satisfaction with all their reported dogs. Ninety percent (*n* = 777/859) of females and 89% (*n* = 66/74) of males indicated satisfaction with at least one dog. Seventy-six percent (*n* = 649/859) of females and 78% (*n* = 58/74) of males indicated satisfaction with all their reported dogs.

Ninety-one percent (*n* = 851/933) of participating owners indicated they would consider another dog in the future. Ninety-one percent (*n* = 644/707) of owners who were satisfied with all their dogs indicated they would consider another dog in the future; three percent (*n* = 44/707) indicated they would not. Ninety-six percent (*n* = 131/136) of owners that indicated satisfaction with some, but not all, of their dogs indicated they would consider another dog in the future; three percent (*n* = 4/136) indicated they would not. Eighty-four percent (*n* = 76/90) of owners who were not satisfied with any of their dogs indicated they would consider another dog in the future; 14% (*n* = 13/90) indicated they would not.

Owners indicated that their expectations had been met for 100% (*n* = 1/1) of the dogs that had been lost, 94% (*n* = 32/34) of the dogs that had been euthanized, 86% (*n* = 6/7) of the dogs not living with their owner for reasons other than those asked on the questionnaire, 83% (*n* = 118/142) of the dogs that had died, 20% (*n* = 1/5) of the dogs that had been rehomed, and none (*n* = 0/1) of the dogs that had been surrendered.

### 3.7. Impact of Adoption Criteria on Owner Satisfaction

#### 3.7.1. Additional Preprocessing

Two primary motivation categories (protection and social interaction) and two adoption source categories (found and imported from a foreign country or island) were dropped from all models due to violations of the “rule of ten” [34].

#### 3.7.2. Models Accounting for Characteristic Ranks

According to the model fit to the data using personality rank in addition to primary motivation, adoption source, background variables, and intercept, a one-unit deprioritization in rank for personality (i.e., a move from 1st place to 2nd place) resulted in 0.86 (95% CI: 0.79–0.94; *p* = 0.001) the odds of indicating satisfaction with an adoption. Owners who indicated primary motivations of an exercise partner (OR: 0.45; 95% CI: 0.25–0.82; *p* = 0.014), requests from another family member (OR: 0.29; 95% CI: 0.16–0.53; *p* < 0.001), for working/sporting (OR: 0.47; 95% CI: 0.26–0.87; *p* = 0.022), or some unspecified reason (OR: 0.40; 95% CI: 0.26–0.62; *p* < 0.001) had decreased odds of indicating satisfaction with an adoption compared to those who indicated companionship as their primary motivation. In addition, owners with greater than six years of forethought (OR: 0.25; 95% CI: 0.13–0.56; *p* = 0.001) or greater than six months but less than six years of forethought (OR: 0.51; 95% CI: 0.31–0.84; *p* = 0.023) had decreased odds of indicating satisfaction with an adoption compared to those with less than one week of forethought. No other independent variable was found to have a significant correlation with the outcome due to either exceeding the study significance level or confidence interval disqualification.

No other model accounting for characteristic rank was found to provide a better fit to the data than the model accounting for primary motivation. A hierarchical overview of the models used in this study is provided in Table 2.

## 4. Discussion

Over half of the dogs in this study were acquired from a shelter. Participants who acquired a dog from a rescue or shelter had an average age of 51 years, which is interesting since other studies exploring social and ethical methods of acquisition found a high percentage of adopters in the millennial age group [8,40]. A study that employed probability-based panel to generate a nationally representative sample of adults in the United States yielded a similar average participant age of 48 years (range: 18 to 94 years old) [41]. The majority of dog owners (92%) responding to the survey were female, which is very typical of these types of surveys [42,43,44,45].

A high percentage (83%) of participants were fully satisfied with their choice of dog with less than one percent being dissatisfied. A possible reason for this favorable outcome may have been population bias; participants in this study were a group of dedicated dog owners who chose to engage in our online platform in their spare time. Additionally, we cannot rule out the possibility that dog owners may be less inclined to share their adoption experiences when none of their adopted dogs have met their expectations. An even higher percentage of owners (95%) whose dogs had been euthanized indicated that their expectations had been met. One likely explanation for this slightly different finding is that these dogs were euthanized for medical reasons. Logically, owner satisfaction was much lower for dogs that had been rehomed or surrendered (20% and 0%, respectively).

Various selection criteria that we presented as possibly relevant to prospective dog owners, the dog’s personality was the only one found to have a significant positive effect on eventual owner satisfaction. This is surprising since determining personality is not easy even for pet professionals [46] and, in addition, shelters and rescues are typically loud and visually over-stimulating for dogs making selection based on a dog’s personality even more difficult [47]. A dog in this situation will likely be over-exuberant or displaying fearful behavior [48]. None of the other selection criteria we suggested, including age, appearance, breed, compatibility with other pets, size, or trainability, was found to have a significant effect on eventual owner satisfaction. Neither did we find breed to be an adoption criterion that led to greater or less eventual owner satisfaction. This is in distinction from the results of a study by Posage et al. [49] who found that certain breeds, including toys and terriers, were more successfully adopted.

Although Garrison and Weiss [50] found that no particular adoption criterion drove the dog adoption process, we found the dog’s personality was, on average, the most highly prioritized factor in adoption. As far as future adoptions were concerned, our participants were more likely to prioritize behavioral elements such as the dog’s personality, compatibility with other pets, and trainability. Conversely, participants were more likely to deprioritize the ranking of physical attributes such as age, appearance, and breed for future adoptions.

Compared to owners who adopted dogs primarily for companionship, those who selected their dog as an exercise partner, at the request of another family member, for working/sporting purposes, or some unspecified reason were less likely to indicate eventual satisfaction with their adoption.

Owners with six or more months forethought had decreased odds of satisfaction with an adoption than those with less than one week of forethought. In other words, a lengthy thought process about desired adoption criteria has a detrimental effect on eventual owner satisfaction, possibly because such owners have fastidious requirements for their future charge. On the other hand, owners who already know what they are looking for in a dog to be adopted would be expected to make a more rapid decision and to fare much better because of their definitive and clear-cut selection criteria. It is also possible that youthful and therefore behaviorally malleable dogs were the ones adopted more quickly and integrated more successfully. In support of this contention, Normando et al. [51] found that “young age” was the most important factor leading to quick pet dog adoption. Another possible explanation is that older dogs, a known and relatively immutable commodity, were more rapidly and successfully adopted. Unfortunately, we did not collect data regarding dog age at the time of adoption in this study; therefore, we are unable to formally verify the age group of dogs at the time of adoption as an explanation for the success of rapid adoptions.

The results of this study indicate that potential adopters should seriously consider the personality and behavior of a dog rather than breed and size. Shelters would be well advised to take this to heart and advise potential adopters accordingly.

## 5. Conclusions

While our primary interest was to identify key characteristics that adopters use to select dogs and how satisfied they were with the result of their selection, we found the dog’s personality/behavior as the only characteristic with a positive effect on eventual owner satisfaction. Less forethought, ideally less than one week, was found to have a positive effect on eventual owner satisfaction. Adopting a dog for companionship and affection was more likely to lead to eventual owner satisfaction than any other motive. In our study, the greatest degree of owner satisfaction was assessed in dogs that were eventually euthanized, possibly because euthanasia was performed at the end of the dog’s life and/or for medical reasons. In comparison to the selection criteria priorities of prior adoptions, the selection criteria rankings that participants had indicated they would employ for future adoptions tended to shift away from physical to behavior characteristics.

## Figures and Tables

**Figure 1 animals-12-02264-f001:**
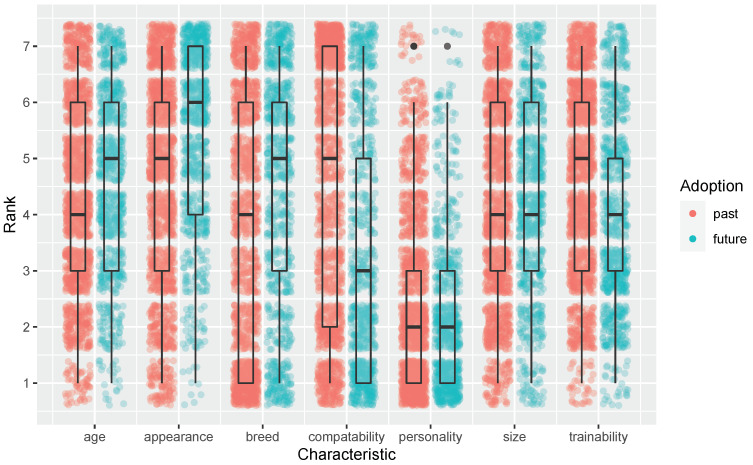
Jitter plot of characteristic rank positions for past and future adoptions. Box plots are overlaid to indicate quartiles and mean.

**Table 1 animals-12-02264-t001:** Average characteristic rank positions for past and future adoptions. A lower rank position indicates a higher priority characteristic.

Characteristic	Average Rank Position ($\bar{x}$)		$\Delta \bar{x}$	*p*
	Past Adoptions	Future Adoptions		
Age	4.31	4.59	0.28	<0.001
Appearance	4.54	5.47	0.93	<0.001
Breed	3.63	4.38	0.75	<0.001
Compatibility	4.49	3.33	−1.16	<0.001
Personality	2.5	2.16	−0.34	<0.001
Size	4.28	4.3	0.02	0.787
Trainability	4.65	4.01	−0.64	<0.001

**Table 2 animals-12-02264-t002:** Hierarchical regression analysis of adoption criteria on owner satisfaction.

Predictor	Logistic Regression Coefficient (β)									
	Model 1	Model 2	Model 3	Model 4a	Model 4b	Model 4c	Model 4d	Model 4e	Model 4f	Model 4g
Step 1. Background										
Owner is male	0.01	0.19	0.18	0.18	0.20	0.18	0.17	0.23	0.17	−0.18
Owner age (in years)	0.00	0.00	0.00	0.00	0.00	0.00	0.00	0.00	0.00	0.00
Forethought (ref: <1 week)										
1 week to 6 months	−0.19	−0.27	−0.47	−0.46	−0.49	−0.46	−0.47	−0.49	−0.47	−0.47
>6 months to 6 years	−0.12	−0.32	−0.59 *	−0.59 *	−0.63 *	−0.59 *	−0.60 *	−0.67 *	−0.60 *	−0.59 *
>6 years	−0.75	−1.00 *	−1.35 **	−1.34 **	−1.38 **	−1.34 **	−1.38 **	−1.38 **	−1.37 **	−1.34 **
Step 2. Adoption Source (ref: pet store)										
Breeder		0.42	0.42	0.41	0.37	0.45	0.42	0.46	0.44	0.42
Family/friend		−0.37	−0.32	−0.32	−0.35	−0.32	−0.30	−0.31	−0.31	−0.32
Shelter/rescue		−0.25	−0.33	−0.33	−0.37	−0.34	−0.31	−0.40	−0.33	−0.33
Step 3. Motivation (ref: companionship)										
Companion for another pet			−0.51 *	−0.52 *	−0.55 *	−0.52 *	−0.44	−0.44	−0.51 *	−0.51 *
Exercise			−0.82 *	−0.82 *	−0.85 **	−0.82 *	−0.82 *	−0.80 *	−0.81 *	−0.81 *
Requested by Family			−1.27 ***	−1.27 ***	−1.28 ***	−1.28 ***	−1.26 ***	−1.25 ***	−1.28 ***	−1.27 ***
Working/sporting			−0.74 *	−0.74 *	−0.79 *	−0.74 *	−0.73 *	−0.75 *	−0.72 *	−0.74 *
Other			−0.90 ***	−0.90 ***	−0.94 ***	−0.91 ***	−0.85 ***	−0.91 ***	−0.89 ***	−0.90 ***
Step 4. Characteristics Rankings										
a. Age				0.02						
b. Appearance					0.07					
c. Breed						0.02				
d. Compatability							0.03			
e. Personality								−0.15 **		
f. Size									−0.02	
g. Trainability										0.00
**Metric**										
Δdf	5	3	5	1	1	1	1	1	1	1
$L^{2}\;\left(\chi^{2}\right)$	4.81	17.14 ***	35.42 ***	0.21	3.18	0.20	0.87	12.08 ***	0.38	0.01
$R_{nk}^{2}$	0.01	0.02	0.06	0.06	0.07	0.06	0.06	0.08	0.06	0.06
$H-L\;(\chi^{2})$	0.01	0.02	0.06	0.06	0.07	0.06	0.06	0.08	0.06	0.06

* = *p* ≤ 0.05; ** = *p* ≤ 0.01; *** = *p* ≤ 0.001.

## Data Availability

The data presented in this study are openly available in Zenodo at 10.5281/zenodo.6910622, reference number [52].

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
