# Peer review of "Selection Factors Influencing Eventual Owner Satisfaction about Pet Dog Adoption"

_animals, 2022, doi:10.3390/ani12172264_

Round 1

Reviewer 1 Report

This study investigates factors that influence dog owners’ satisfaction post-adoption. This is an important topic and would be very useful to adoption facility. The research design and analysis are robust with some interesting research finding. I really like Figure 1 that beautifully demonstrates the change of characteristics valued by owners pre and post adoption. Overall, this is a good paper and I just have a few comments as below.

Line 50: “Shore, 2005” Please fix the reference format.

Line 80-103: Thank you for providing the details of the questionnaire. It would be great if the questionnaire, including all questions, can be presented in a table and put it in the appendix.

Line 94-95: “The characteristics that owners were … compatability … and trainability”. “Compatability” sounds vague to me as it can encompass many different factors including other characteristic options (e.g. personality and trainability). Please define the term.

Data processing: I presume the data was tidied using R. If yes, please mention it in the manuscript, including what version of R was used.

In this study, adoption sources included breeder, online pet shop, and local pet shop, etc. This might sound silly but I was a little bit unsure. Would authors please define what does “adoption” mean in this study? Does “adoption” here involve purchasing a dog from breeders and pet shops?

Line 242: Please explain or provide a reference about “rule of ten” for readers who are not familiar with the statistical jargon.

Table 2: I would encourage authors to run another hierarchical regression analysis using the same input variables (adoption criteria) but replace the output with whether the dog lived or did not live with the owner, to investigate the direct relationship between adoption criteria and successful adoption. Afterall, owners being satisfied with their dogs does not necessarily mean that they would still keep the dog. However, due to the low number of unsuccessful adoption (e.g. dogs being rehomed and surrendered), this might not be achievable in this study. Also, because of this, I would suggest the title being changed to something like “Factors Influencing Eventual Owner Satisfaction about Pet Dog adoption”, because there was no direct link between factors and adoption decision.

Line 267-274: To determine the representative of this data, what is the rate of successful adoption in the same time period across the US (or perhaps the similar region)? It might be that the majority of people completing this survey were those with successful adoption while those returned their dogs did not want to participate in the study, and this might create a bias that need to be addressed.

Discussion: I would encourage to add a short section of authors’ suggestions to adoption facility based on the finding of this study. For example, how adoption facility can modify their current practice to improve the successful adoption. Also, it will be useful to discuss how this study can be used to educate first time potential owners, many of whom still prioritize the physical appearance of a dog.

Reviewer 2 Report

Review of manuscript animals-1860938, Factors Influencing Adoption of Pet Dogs and Eventual Owner Satisfaction by Dinwoodie, Zottola, Jubitz, and Dodman.

The authors address an important topic – factors that might influence how satisfied a person is with an adopted dog.  The authors argue that being satisfied with an adopted dog will decrease the likelihood that the animal is returned.  One of the main conclusions is that people who prioritize the dog's personality were likely to be satisfied with the dog.

This conclusion is not new.  Rather than just asking about the importance of personality in adopting a dog, Curb et al. (2015) investigated eight dimensions of a dog's personality and how they influence satisfaction with a dog.  Four dimensions were found to be related to satisfaction.  Thus, one of the main conclusions of the current manuscript could be predicted from the literature with more detail than the current manuscript can provide.   The authors do not cite this literature in their introduction.

Other literature that might be appropriate to include in the introduction include van Herwijnen et al. (2018) which addresses owner satisfaction with a dog and Meyer and Forkman (2014) which address the dog-owner relationship which should be related to satisfaction.  Neither are included in the introduction.

My main methodological concern with the manuscript is that the sample is likely biased with 92% of the dog owners who responded to the survey being female (line 172).  The mean age of the dog owners who responded to the survey also might be higher than in the population.  If the sample does not represent the population, how can you generalize the results to the population?  Minimally, this must be addressed in the discussion.  To maximize your contribution to the literature, a more representative sample must be used.

A minor methodological concern is that the authors provided only two option for the participant's gender – female and male.  With a sample size of 971, I would be very surprised if all of the participants fit into one of these two categories.  I encourage the authors to be more inclusive in their gender options in future surveys.  Doing so will likely lead to a more representative sample as some people who do not fit into the traditional categories are likely to stop responding to the questionnaire when their gender is not listed.

Another minor issue is that the research is based on a survey which provides self-reported data which can be different from objective measurements of the same variable.  While it is cumbersome to repeatedly say, for example, "self-reported satisfaction", this distinction needs to be at least mentioned.

Lines 223-227: How much of this is a bias in those who participated in the study.  Are people who are more satisfied with their dogs more likely to be interested in dogs in general, look for information about dogs on the internet, and to find your study?  Are women more likely to report being satisfied than men, perhaps due to a stereotypical nurturing role played by women?

Lines 240-242: How many observations were dropped for these reasons?

Minor comments:

Line 8: "Odds of eventual satisfaction higher" is awkward.  Consider "Odds of eventual satisfaction are higher"

Line 45: "therefore returns more likely" is awkward.  Consider 'therefore returns are more likely."

Lines 47- 48:  I am not sure what you mean by "Dog-human personality".  Is this the personality of the dog, the personality of the human, the interaction of the dog's personality and the human's personality, or something else?

Line 48: "as for example" is awkward.  Consider ", for example, "

Line 48:  A noun or pronoun is needed before "found it alone…".  Consider "Walker et al. found that it alone…".

Line 50: Ditto line 48.

Lines 52-54:  Provide a citation to support this statement of fact.

Line 57: "hours to days" is not a rate.  Consider "Other studies found return periods as short as hours to days"

 Line 58: Replace "Returning shelter dogs" with "Whether an adopter returns a dog to a shelter"

Line 58: Insert "the" between "on" and "severity".

Line 60: Consider replacing "dissolve" with "harm"

Line 63: Consider replacing "better management" with "improve management"

Line 64: Insert "the" before "Center for Canine"

Line 69: Change "none was" to "none were"

Line 110: "Data" is a plural noun and requires a plural verb.

Line 163: Replace "Psuedo" with "Pseudo"

In Table 1, "Δμ" appears to mean the difference between the sample means of the past adoption rank and the future adoption rank.  However, μ implies a population mean but the table contains sample means.

Line 233: Insert "who were" before "not satisfied"

Line 276:  Consider inserting "the dog's" before "personality".  While you did not look at it, the human's personality and the interaction between the human's and the dog's personalities influence satisfaction with the dog.

Lines 288, 290, 314: Consider inserting "the dog's" before "personality".

References

Curb, L. A., Abramson, Ci, Grice, J. W., & Kennison, S. M. (2015).  The relationship between personality match and pet satisfaction among dog owners.  Anthrozoös, 26(3), 395-404.  https://doi.org/10.2752/175303713X13697429463673

Meyer, I. & Forkman, B. (2014).  Dog and owner characteristics affecting the dog-owner relationship.  Journal of Veterinary Behavior, 9(4), 143-150.  https://doi.org/10.1016/j.jveb.2014.03.002

van Herwijnen, I. R., van der Borg, J. A. M., Naguib, M., & Beerda, B. (2018).  Dog ownership satisfaction determinants in the owner-dog relationship and the dog'b behavior.  PLoS ONE, 13(9), e0204592. https://doi.org/10.1371/journal.pone.0204592

Round 2

Reviewer 2 Report

The authors have adequately addressed all of my concerns about the manuscript.

On line 78, should "adoption" be "adopter"?
